# Geomorphological Model Comparison for Geosites, Utilizing Qualitative–Quantitative Assessment of Geodiversity, Coromandel Peninsula, New Zealand

**Vladyslav Zakharovskyi** [1,*] **and Károly Németh** [1,2,3]

1 School of Agriculture and Environment, Massey University, Palmerston North 4474, New Zealand
2 Institute of Earth Physics and Space Science, 9400 Sopron, Hungary
3 The Geoconservation Trust Aotearoa Pacific, Ōpōtiki 3122, New Zealand
* Correspondence: v.zakharovskyi@massey.ac.nz; Tel.: +64-22-357-4973

**Abstract:** In qualitative–quantitative assessment of geodiversity, geomorphology describes landscape forms suggesting specific locations as geosites. However, all digital elevation models (DEM) contain information only about altitude and coordinate systems, which are not enough data for inclusion assessments. To overcome this, researchers may transform altitude parameters into a range of different models such as slope, aspect, plan, and profile curvature. More complex models such as Geomorphon or Topographic Position Index (TPI) may be used to build visualizations of landscapes. All these models are rarely used together, but rather separately for specific purposes—for example, aspect may be used in soil science and agriculture, while slope is considered useful for geology and topography. Therefore, a qualitative–quantitative assessment of geodiversity has been developed to recognize possible geosite locations and simplify their search through field observation and further description. The Coromandel Peninsula have been chosen as an area of study due to landscape diversity formed by Miocene–Pleistocene volcanism which evolved on a basement of Jurassic Greywacke and has become surrounded and partially covered by Quaternary sediments. Hence, this research provides a comparison of six different models for geomorphological assessment. Models are based on DEM with surface irregularities in locations with distinct elevation differences, which can be considered geosites. These models have been separated according to their parameters of representations: numerical value and types of landscape. Numerical value (starting at 0, applied to the area of study) models are based on slope, ruggedness, roughness, and total curvature. Meanwhile, Geomorphon and TPI are landscape parameters, which define different types of relief ranging from stream valleys and hills to mountain ranges. However, using landscape parameters requires additional evaluation, unlike numerical value models. In conclusion, we describe six models used to calculate a range of values which can be used for geodiversity assessment, and to highlight potential geodiversity hotspots. Subsequently, all models are compared with each other to identify differences between them. Finally, we outline the advantages and shortcomings of the models for performing qualitative–quantitative assessments.

**Keywords:** GIS modeling; Geomorphon; Topographic Position Index; ruggedness; roughness; total curvature; slope; geoconservation; geotourism



## 1. Introduction

A variety of geodiversity assessment methods have been applied by geoscientists to areas of study to highlight locations which may contain high-value features based on abiotic factors. The range of abiotic elements (geology, geomorphology, hydrology, climate, cultural heritage, soils, and others) are compounded to provide a full description of the geodiversity of different studied areas throughout the Earth [1–7]. However, at a global scale, many areas do not display enough surface features or qualities to provide a useful reflection of historical and prehistorical events. Understanding landscape evolution is one of the most important research projects in geology, as it provides a range of possibilities

for science development, especially for protection against hazard events, conservation, tourism, and education. Tectonic and erosion processes are the main drivers of surface and crustal evolution, creating new geological forms through the tilting and uplifting of old formations once buried underground; creation, transportation, and deposition of sedimentary particles; and creation of new outcrops. New formations and exposed surface areas provide information that describes the history of the Earth's evolution. Patterns and cycles of rock alteration are predominantly led by the geographical–geological cycle [8–10] of orogenetic uplift and volcanism, subsequently acted on by weathering and erosion [4,11]. The abiotic environment is shaped by two main elements: geology (parameter of quality or mineral (element) composition) and geomorphology (forms of rocks created after exogenic and endogenic processes). These processes are captured in the cycle of alterations which act over thousands of years. Hence, these elements are the most important part of geodiversity assessment, especially in research directed toward an understanding of the Earth's history. Moreover, geosites may contain this specific information, with their recognition a starting point for further field observations. Therefore, we define geosites as those sites which best represent the concepts of geosystems that we have outlined.

Geographical information systems (GIS) provide a range of tools, which can be utilized for geodiversity description and/or geosite recognition based on the stated aim of research and data availability. Our research aims to demonstrate geodiversity assessment, using six geomorphological models based on the same source for the data set. These models are slope angle, roughness, ruggedness, and total curvature, while the Topographic Position Index (TPI) and Geomorphon are more complex models applied to different types of landscape. All models have been generated from the Shuttle Reader Topography Mission (SRTM) data set [12] as it represents freely accessible data, which allow us to simplify calculations and describe methodologies easily reproduced by other researchers.

Qualitative–quantitative assessments of geodiversity are based on arithmetic average values between geological and geomorphological elements, described by a seven-point system [13], which has been changed in this assessment to a better tailored system for our study area (more details in Section 2. Material and Methods). Six geomorphological models have been assessed based on geological elements to create a general geodiversity model of the region. Geodiversity models are then compared in how they recognize geosites. Geological elements are evaluated by a seven-point range of values, based on global rarity of rock type found exposed on the surface. However, a detailed evaluation of numerical geomorphological models (slope, ruggedness, roughness, and total curvature) is beyond the scope of this paper, though they have been included for comparison through straight multiplication with geological values. Meanwhile, landscape models (TPI and Geomorphon) have been evaluated based on expert views.

The Coromandel Peninsula in the North Island of New Zealand is considered a good study area for the assessment of geosite recognition. This area contains several different rock types, creating unique geological and geomorphological formations, including mountain ranges, meadow hills, plains, and coastal cliffs [14,15]. It is significant that the area contains some of the earliest sites of Māori and then European settlements, geological resources, and is a globally recognized tourist destination [16–21]. Therefore, our research is relevant for the establishment of geoeducation, geoconservation, and/or geotouristic projects.

The main aim of our research is to compare TPI, Geomorphon, slope, roughness, ruggedness, and total curvature models as these are the most influential geomorphological elements for assessment of geodiversity. Moreover, the methodology must be simple and repeatable for other users and applicable for different territories throughout the world. Additionally, the results of our research will assist in demonstrating, understanding, and describing the differences and similarities between territories. This will be valuable for qualitative–quantitative assessments of geodiversity, which can be used to accurately locate potential geosites.

## 2. Materials and Methods

### 2.1. Sample

The Coromandel Peninsula is located on the northeast side of North Island, New Zealand. It comprises a territory approximately 40 km wide and 100 km long with a NW–SE orientation. The peninsula is contiguous with the Bay of Plenty on the southeast, Hauraki Gulf on the southwest, and the eastern shores open to the Pacific Ocean (Figure 1) [14,15]. We selected the peninsula as an area of research for this assessment as the region contains diverse biological and geological units and forms and contains significant conservation reserves and tourist destinations. Moreover, geological diversity of the territory is shaped by volcanic interaction with marine–sedimentary environments during the Miocene–Pleistocene. Evolution of features can be recognized throughout the research area, creating different types of relief, from mountain ranges and meadow hills to marshes and plains. We suggest that such a high amount of geological and geomorphological diversity provides potential opportunities for landscape evolution as a basis for education and community engagement. Additionally, it was the settlement region for the first Māori tribes, which leaves an important cultural footprint for anthropological studies, while Europeans used this region for mining based on the gold and silver epithermal deposits [22–24]. Hence, our geomorphological assessment of the Coromandel Peninsula will provide a firm foundation for future research based on tourism, education, and conservation development.

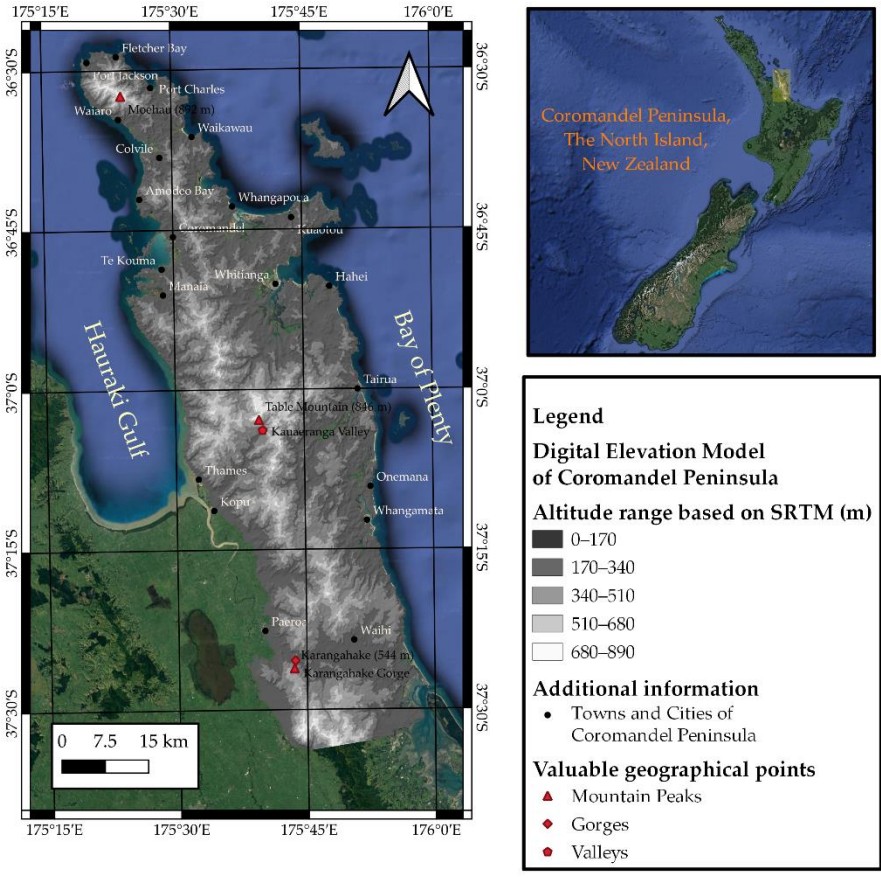

**Figure 1.** Overview map and elevation model of Coromandel Peninsula created from Shuttle Radar Topography Mission (SRTM) 1 Arc-Second Global (https://www.usgs.gov/centers/eros/science/usgs-eros-archive-digital-elevation-shuttle-radar-topography-mission-srtm-1, accessed on 27 August 2022); background is Google terrain map. The coordinate system is WGS 84 (EPSG: 4327); the same applies for all other figures.

## 2.2. Geological Description

The geology of the Coromandel Peninsula has been formed by Miocene–Pleistocene land-based volcanic activities some Holocene tephra derived from outside of the region. Basement lithologies are formed by the Jurassic siliciclastic rock Greywacke (Figure 2) [14,15,25–30]. Younger clastic sediments can be found mostly on the boundaries of the Coromandel Peninsula, with Greywacke forming the most frequently encountered and largest sediment formation presented in the west and north part [31,32]. Lithologies encountered throughout the region in valleys, depressions, and hollows include mudstone, sandstone, conglomerates, and breccia from the Holocene and Pleistocene periods. Felsic extrusive rocks from the Whitianga Group are represented by rhyolite and ignimbrite spread from the central to the eastern part of the peninsula. The most extensive lithostratigraphic formation on the peninsula is the Coromandel Group, which includes intermediate extrusive andesite and intrusive diorite (granite–granodiorite), also known as "*Coromandel Granite*" [14,15,33–35]. Granodiorite is found at the far northwest part, while andesite mostly forms the whole peninsula and is widely spread from the south to the north. Basalt of the Neogene period is the rarest type of rock in the Coromandel Peninsula, which is exposed on the surface near coastal areas in the northeast part of the peninsula (closest to Great Mercury Island). Finally, a tuff formation can be observed in the transition zone between the far north and central parts of the Coromandel Peninsula. In conclusion, the geological variety of the Coromandel Peninsula is represented by a wide range of volcanic and sedimentary rocks spanning the time frame from the Jurassic to the Holocene periods.

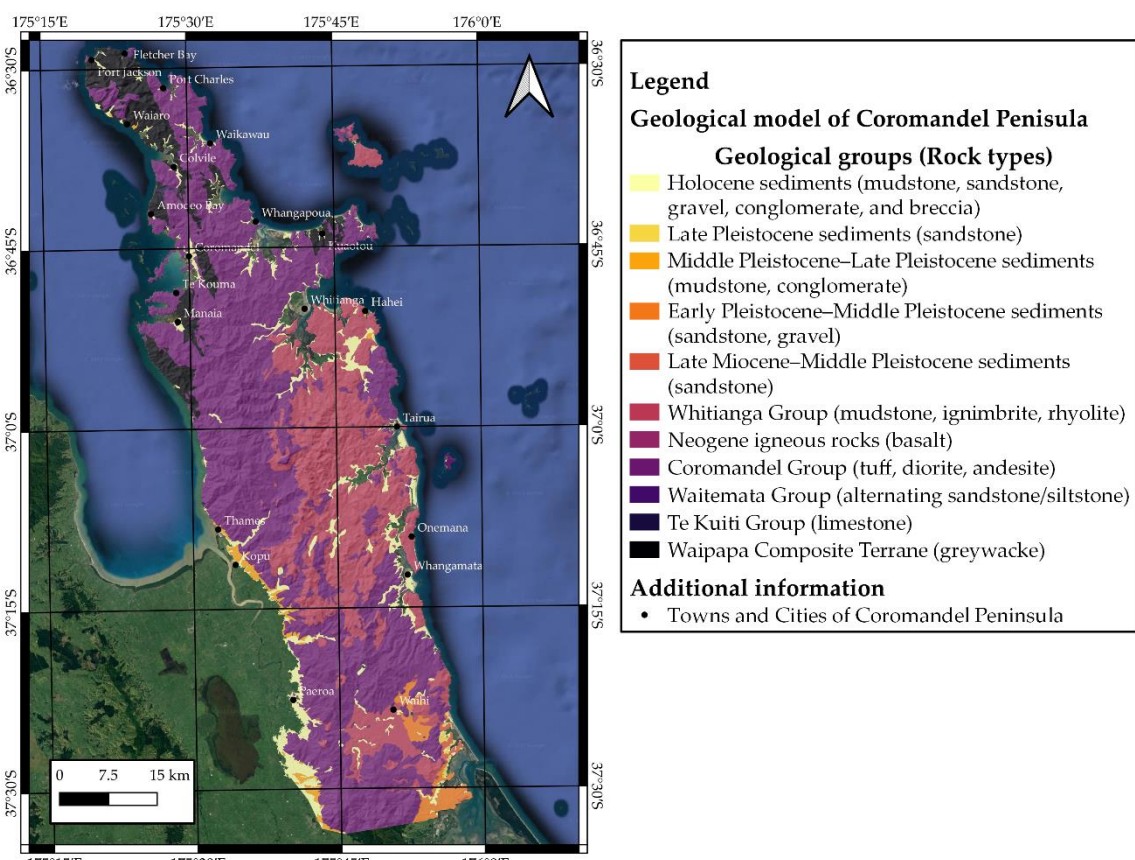

**Figure 2.** Geological model of Coromandel Peninsula based on the 1:250,000 scale New Zealand Geological Map (Q-Map Series—https://www.gns.cri.nz/data-and-resources/geological-map-of-new-zealand/, accessed on 20 August 2022) [36]; background is Google terrain map.

## 2.3. Geomorphological Description

The geomorphology of the studied area is variable based on the elevation model of the Coromandel Peninsula [30]. The landscape of the south is mostly formed by moun-

tain ranges with elevations between 700 and 900 m, mostly rising to the southwest from Karangahake Gorge (Figure 1). The relief is formed by remnants of the Waihi Caldera, with the area becoming flatter closer to sea level as one moves north. A long mountainous ridge at the center of the region commences at the Wharekawa Caldera (near to the settlement of Whangamata) and extends to Colville in the far north, where elevation once more decreases closer to sea level. Moving west to east, a rise in elevation commences in the west, with the highest central point found at Upper Kauaeranga in the central region (Figure 1). Further east landscape features are formed by remnants of the Kapowai Caldera and the Whitianga Volcanic Center. The north part of the peninsula is mostly formed by rolling hills with lower elevations between 180 and 540 m above sea level. However, at Waiaro, elevation rapidly starts to rise from 180 to nearly 900 m, with the highest point being Mount Moehau at 892 m. In conclusion, the territory of the Coromandel Peninsula contains many uneven surfaces with rapid changes in elevation ranging from sea level to 900 m and above, resulting in a high diversity suitable for testing for geomorphological assessment.

*2.4. Methodology*

In this article, we compare six different methods using parameters based on the digital elevation model (DEM) of the Coromandel Peninsula. The methodology is originally based on slope steepness and evaluation of geological units [4,11,13]. The main aim of our qualitative–quantitative assessments is to recognize geosites based on simple calculations of numeric averages; the qualitative aspect of our system is the evaluation system itself, while the quantitative aspect is based on the two main elements of geodiversity (geology and geomorphology) [4]. Geology contains information about qualitative properties of the region, while geomorphology is a measure of its form. However, the main goal of this research is to compare six geomorphological models based on different parameters, with some featuring an axis displaying numerical values from 0 to positive only, while others display growth in the negative direction as well, to describe some specific morphological forms. Hence, the evaluation system cannot be easily applied to each model. Our suggested solution is calculation of parameters based on numerical models (slope, roughness, ruggedness, and total curvature) using multiplication with evaluated geological models (seven-point system as per sub-Section 2.5. Geological Evaluation System). Meanwhile, Geomorphon and TPI as landscape models have been evaluated as demonstrated below (Table 1). The multiplication algorithm has been utilized to create a wider difference between results and does not require separate evaluation of each geomorphological model.

Next, our methodology utilizes a grid formed with 2.5 km length for each side of cells (6.25 km$^2$), applied to the whole Coromandel Peninsula for all six geomorphological models. As the territory of the Coromandel Peninsula is relatively large (~100 km long and 40 km wide), using a non-grid method of scaling will result in poor visibility of most features on the final map [4,13]. Therefore, the 6.25 km$^2$ area was selected as this length aligns with the range of human visibility, meaning that field observations can be applied to the whole cell together with its neighbors [11]. Additionally, the same scale has been used for the creation of topographical and geological standard maps such as the 1:250,000 scale New Zealand Geological Map (Q-Map Series—https://www.gns.cri.nz/data-and-resources/geological-map-of-new-zealand/, accessed on 20 August 2022) [36].

In the assessment, the maximum value of each region has been utilized for calculation of geology and geomorphology, in order not to miss potential geosite locations. This could occur when low values are seen in the surrounding assessed region, except in the case of the Geomorphon model. Therefore, every region shows potential for containing valuable sites, which can then be further defined through more precise observations, for example, on-ground field work. However, this methodology is of no use for describing geodiversity description as it avoids low-value locations. In contrast, the Geomorphon model based on default parameters will suggest small features throughout the cell, thereby resulting in an entirely homogeneous research area. Hence, Geomorphon is modeled on an arithmetic average calculated for each cell (Table 1).

**Table 1.** The value systems for geodiversity assessment of Coromandel Peninsula.

| Values (7-Point System) | Main Values of Geodiversity | | | |
|---|---|---|---|---|
| | Elements of Geodiversity | | | |
| | Geomorphology | | | Geology |
| | Slope, Roughness, Ruggedness, Total Curvature | Topographic Position Index | Geomorphon | Rock Type and Ages |
| 1 (the lowest) | The numerical models have been included into assessment without direct evaluation. | Topographic Position Index model have been evaluated by 7-point system for positive and negative forms of landscape, where 0 is the lowest value | Flat and Slope | Sedimentary Cenozoic |
| 2 (low) | | | Hollow and Spur | Sedimentary Mesozoic |
| 3 (low to middle) | | | Footslope and Shoulder | Sedimentary Paleozoic |
| 4 (middle) | | | Valley and Ridge | Metamorphic Precambrian |
| 5 (middle to high) | | | Depression and Summit | Intrusive Precambrian |
| 6 (high) | | | 5-point system | Extrusive Cenozoic |
| 7 (the highest) | | | | Extrusive Mesozoic |
| 8 (the rarest) Only Rocks | Sedimentary (Precambrian), Metamorphic and Intrusive (Cenozoic, Mesozoic, Paleozoic), Extrusive (Paleozoic, Precambrian) | | | |

In this research, we utilized Shuttle Radar Topography Mission (SRTM) 1 Arc-Second Global (https://www.usgs.gov/centers/eros/science/usgs-eros-archive-digital-elevation-shuttle-radar-topography-mission-srtm-1, accessed on 27 August 2022) for DEM. Access is free and data are available for the entire surface of the Earth at a resolution of 30 m per pixel, which we consider suitable for large-scale geodiversity calculations. Then, QGIS (3.16 "Hannover") (https://qgis.org/en/site/forusers/download.html, accessed on 13 August 2022), with its plugin "SRTM-Downloader" (https://plugins.qgis.org/plugins/SRTM-Downloader/, accessed on 15 August 2022), was utilized for model calculation as it contains all the required tools for geomorphological modeling. Hence, six models for geomorphological description were created from SRTM data using QGIS software, based on two main classes, numerical values, and landscape types. Numerical classes are models that have parameters from 0 to the highest value found in research area based on qualities such as slope, ruggedness, roughness, and total curvature. Meanwhile, landscape types are represented by numbers, describing qualities of relief such as slope, plain, valley, or range. These models are the TPI and Geomorphon (Table 1).

*2.5. Geological Evaluation System*

Geological evaluation systems have been created in previous research on qualitative–quantitative assessment for geosite recognition [13]. Currently, the system is for global scale of assessment as it is based on the rareness of different rock types throughout the Earth's surface. Based on the research of Blatt, H. and Jones, R. L. "*Proportions of exposed igneous, metamorphic, and sedimentary rocks*" [37], rock formations were divided into types such as intrusive, extrusive, metamorphic, and sedimentary; and their ages, i.e., Cenozoic, Mesozoic, Paleozoic, and Precambrian (Table 2). All types of rocks and their ages were considered and given values from 1 to 7, with value 8 reserved for only the rarest types, which are less than 1% throughout the global surface (Table 1). The lower values are as follows: 1 point for Cenozoic Sedimentary, the most common type of formation at more than 60%; 2 points are assigned to Mesozoic Sedimentary rocks, occurring at half the amount of the former type; 3 points are assigned to Paleozoic Sedimentary; 4 points are assigned to Precambrian Metamorphic which, although it has a slightly higher extent than the previous rock type (Paleozoic Sedimentary), is given a higher value because metamorphic processes are more complex and may provide more information about geological processes; a value of

5 points is assigned to Precambrian Intrusive; and finally, Cenozoic and Mesozoic Extrusive rocks are given 6 and 7 points, respectively. The remaining rock types were grouped under 8 points because of their rareness: Precambrian Sedimentary; Cenozoic, Mesozoic, and Paleozoic Metamorphic and Intrusive; Paleozoic and Precambrian Extrusive. Using this scale, it is apparent that the Coromandel Peninsula contains many formations that can be given a value of 6 for rareness, represented by extrusive rocks of Cenozoic time (Andesite, Rhyolite, Ignimbrite, Dacite); while Greywacke is given a value of 2 points (Mesozoic Sediments). Quaternary sediments found throughout the Peninsula are given 1 point. Finally, "*Coromandel Granite*" is one of the rarest types of Mesozoic Intrusive rocks that can be found at the Earth's surface, so its value is 8.

**Table 2.** Percentage of rock types exposed on the Earth's surface as a function of geological age [37].

| Eras | Crystalline | | | | Sedimentary | No. of Usable Data Points |
|---|---|---|---|---|---|---|
| | **Extrusive** | **Intrusive** | **Metamorphic and "Precambrian"** | **Total** | | |
| Cenozoic | 4 | 0 | 0 | 4 | 33 | 290 |
| Mesozoic | 2 | 1 | 1 | 4 | 18 | 177 |
| Paleozoic | 1 | 1 | <1 | 2 | 13 | 117 |
| Precambrian | 0 | 6 | 15 | 21 | 1 | 173 |
| Age unknown | 1 | 1 | 1 | 3 | 1 | 26 |
| Total | 8 | 9 | 17 | 34 | 66 | 783 |

## *2.6. Geomorphological Evaluation System*

For geomorphological assessments, we considered two types of models: numerical (slope angle, ruggedness, roughness, and total curvature) and landforms (Geomorphon and Topographical Position Index (TPI)). Below, we present a short description of each model. In the Results section, they are compared with each other based on a qualitative–quantitative type of assessment of geodiversity. We utilize a 7-point scale for geological value multiplied by the value of each geomorphological model. A limitation of this assessment is that it is unable to utilize a global evaluation system as models are based on SRTM with a resolution of 30 m for pixel. At this resolution, we are unable to clearly define all high slope areas (especially coastal cliffs) throughout the peninsula; however, we consider it adequate to define the highest-value areas.

### 2.6.1. Slope Model Description

The slope model was calculated utilizing (Saga GIS module) in QGIS named "Slope, aspect, curvature". For assessment, we utilized the default method "9 parameter 2-nd order polynom" created by Zevenbergen and Thorne (1987) (https://onlinelibrary-wiley-com.ezproxy.massey.ac.nz/doi/epdf/10.1002/esp.3290120107, accessed on 21 August 2022) [38], where they modified Evens' quadratic equation. The model has been used in different types of studies, ranging from geology and agriculture to trail and road plannings. Moreover, it has shown good results in wildfire and flood hazardous areas of research [39,40]. For our database, we utilized the SRTM model of the Coromandel Peninsula downloaded through the QGIS plugin "SRTM-downloader". This was modified with the "Gaussian filter" (Saga tool in QGIS) to smooth a surface and applied to every model described below. This resulted in a slope model of the Coromandel Peninsula containing values ranging from 0 to around 46 degrees, where the highest values are mainly found in the central and northern part of the Peninsula.

### 2.6.2. Roughness Model Description

Roughness is a parameter describing the degree of surface irregularity. Topography is the main factor influencing the parameter of roughness, which can also be influenced by altitude and surface features such as trees, buildings, relief, and terrain [41]. Its calculation is based on identifying differences between neighboring cells and pixels describing features in

those cells. This type of modeling is commonly used for river morphology, climatology, and geography (https://docs.qgis.org/2.8/en/docs/user_manual/processing_algs/gdalogr/gdal_analysis/roughness.html, accessed on 21 August 2022). The roughness model was calculated from SRTM data utilizing QGIS software through the module GDAL "Roughness". This results in a model that appears like a slope, but with different parameters ranging from 0 to 63 describing the surface irregularity. Hence, the highest points of the roughness model are mostly found in the central and northern part of the Coromandel Peninsula.

### 2.6.3. Ruggedness Model Description

Ruggedness calculation parameters have been described by Riley et al. (1999) as the quantitative measurement of the differences in terrain (heterogeneity) [42]. Ruggedness has been used for in-habitat modeling to predict types of species habitats, their density, and variety. Then, ruggedness has demonstrated strong results in paleoglacier studies [43,44]. For our calculations, the main parameter was differences in elevation applied to a $3 \times 3$ pixel grid, whereby 8 surrounding cells are compared with the central one. A value of 0 describes a level and even surface, while a higher value describes higher heterogeneity (https://docs.qgis.org/3.4/en/docs/user_manual/processing_algs/qgis/rasterterrainanalysis.html#:~:text=output%20frequency%20distribution-,Ruggedness%20index,the%208%20cells%20surrounding%20it, accessed on 23 August 2022). In QGIS, this parameter has been calculated utilizing the "Terrain Ruggedness Index (TRI)", which is a Terrain Analysis tool (SAGA module). The results of this calculation can also be demonstrated visually, same as those of the roughness and slope models, but the values range from 0 to 72, with the highest points found in the same geographical areas as the former models.

### 2.6.4. Total Curvature Model Description

Total curvature or general curvature is a parameter which combines plan and profile curvatures and is used here for understanding the flow in our studied territory [45]. Total curvature can range from a starting point of 0 and move in a positive or negative direction, describing different types of surface, such as flat, hilly, and dissected by valleys, respectively [45]. For our calculation, we utilized the same tool as for slope calculation named "Slope, aspect, curvature" (Saga GIS module) in QGIS. For our assessment, we utilized the default method "9 parameter 2-nd order polynom" [38]. The results of our calculation have values ranging from 0 to $8.66252 \times 10^{-5}$, where the highest values are mostly concentrated in the southern part of the center of the Coromandel Peninsula and reflecting the positive form of landscape represented by the mountain ranges of Coromandel Group formations (Figure 2).

### 2.6.5. TPI Model Description

The Topographical Position Index shows the differences between the parameter of elevation in a central cell and predetermined mean values of its surrounding cells. Mostly calculated to determine the position of the studied slope, it can also be used to classify standard landforms (https://docs.qgis.org/2.8/en/docs/user_manual/processing_algs/gdalogr/gdal_analysis/tpitopographicpositionindex.html, accessed on 25 August 2022) [45]. The model was calculated utilizing the "Topographic Position Index (TPI)" tool as part of the Terrain Analysis (SAGA) module presented in QGIS software. Our calculations result in a model with parameters ranging from $-2.8$ to $2.9$. These were then evaluated utilizing a 7-point system, where 0 is the starting point and values increase in both directions. The positive values represent hillslopes and mountainous terrain, while negative values describe valleys and hollows. As the aim of this research is geosite recognition, both negative and positive types of landforms have high values for this assessment; therefore, equal importance is given to negative and positive distance from the 0 point.

### 2.6.6. Geomorphon Model Description

Geomorphon is one of the newest methods of calculating divergence from a range of specific landscape forms. This model can be calculated using GRASS GIS "r.geomorphon", which is based on the relationship of the cell of assessment with its 8 closest neighboring cells. Neighboring cells can be placed in three different positions: on the same altitude as the studied cell, higher than the studied cell, or lower than the studied cell. The combination of all positions between the neighbors describes the exact type of the terrain. These types of terrain are divided into 10 forms, i.e., flat, slope, footslope, shoulder, valley, ridge, hollow, spur, pit, and peak [46,47]. The 5-point evaluation system is presented according to difficulty in extracting the form from the territory, with values as follows: ridge, valley, and slope are 1; flat is 2; footslope and shoulder are 3; hollow and spur are 4; pit and peak are 5. However, this system has some shortcomings as it is unable to describe the "geographical evolution" of slope and relief. In addition, at a global scale, this method for constructing a useful model requires changes in parameters for the chosen location in order to recognize all the significant places and decrease the amount of microrelief and noise to make the model readable. Hence, the Geomorphon model is highly dependent on the scale and type of evaluation.

### 3. Results

Qualitative–quantitative assessment of geodiversity was utilized based on a seven-point evaluation system for geological aspects and a free unspecified system for geomorphology, which resulted in 6 different models. Our aim is to recognize potential geosites in our study area, which can then be subject to further research for a more detailed description. The first four models share the same type of information, where the highest values are seen at high-value locations found in our study area. These models are slope angle, roughness, ruggedness, and total curvature (Figures 3 and 4). Meanwhile, two other models express some specific landforms: TPI and Geomorphon (Figures 5 and 6). All models were created utilizing QGIS (3.16 "Hannover") software, while additional calculations were made in Excel to contrast and compare results of evaluation modes: equal interval and natural breaks (Jenks) [48]. The results of the two modes are presented below, where each model has been compared with others based on the same mode.

In our results, model values were subject to equal interval mode calculations (Figures 3 and 5), demonstrating that places with high and the highest values should be considered the most likely to contain potential geosites suitable for further assessment and evaluation.

Slope models express the most diverse results after equal interval mode calculations, as represented in Figure 3. The results confirm that the northern region of the peninsula contains several regions with valuable geological formations such as "*Coromandel Granite*" (the highest value) and Miocene andesite from the Coromandel Group (high values). These areas also contain high slope degrees, based on the model. In addition, some areas with high values were also located in the north, closer to the central region of the peninsula, also formed by the same andesite. Meanwhile, Great Mercury Island contains only two areas of potential significance, as one of the few locations of Neogene basalts. Other areas with high and the highest values are found in the central region at the boundary between two extrusive Cenozoic groups, Coromandel (mostly andesite) and Whitianga (ignimbrite and rhyolite). The southern regions of the Coromandel Peninsula contain some areas of high value located near the Waihi and Karangahake regions. Moreover, the eastern region of the Coromandel Peninsula has some high-value areas close to coastal areas, which are most likely formed by near-vertical cliffs.

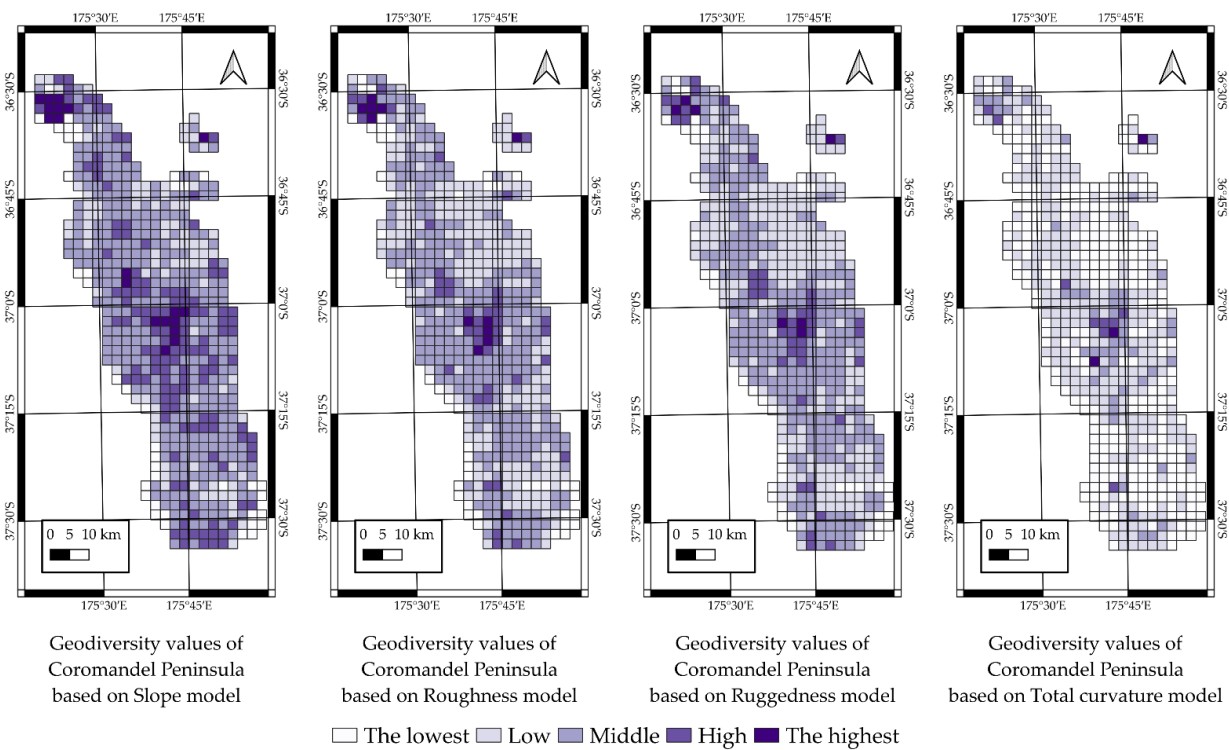

**Figure 3.** Geodiversity values of Coromandel Peninsula for geosite recognition, based on numerical geomorphological models: slope, roughness, ruggedness, and total curvature. Equal interval mode of evaluation.

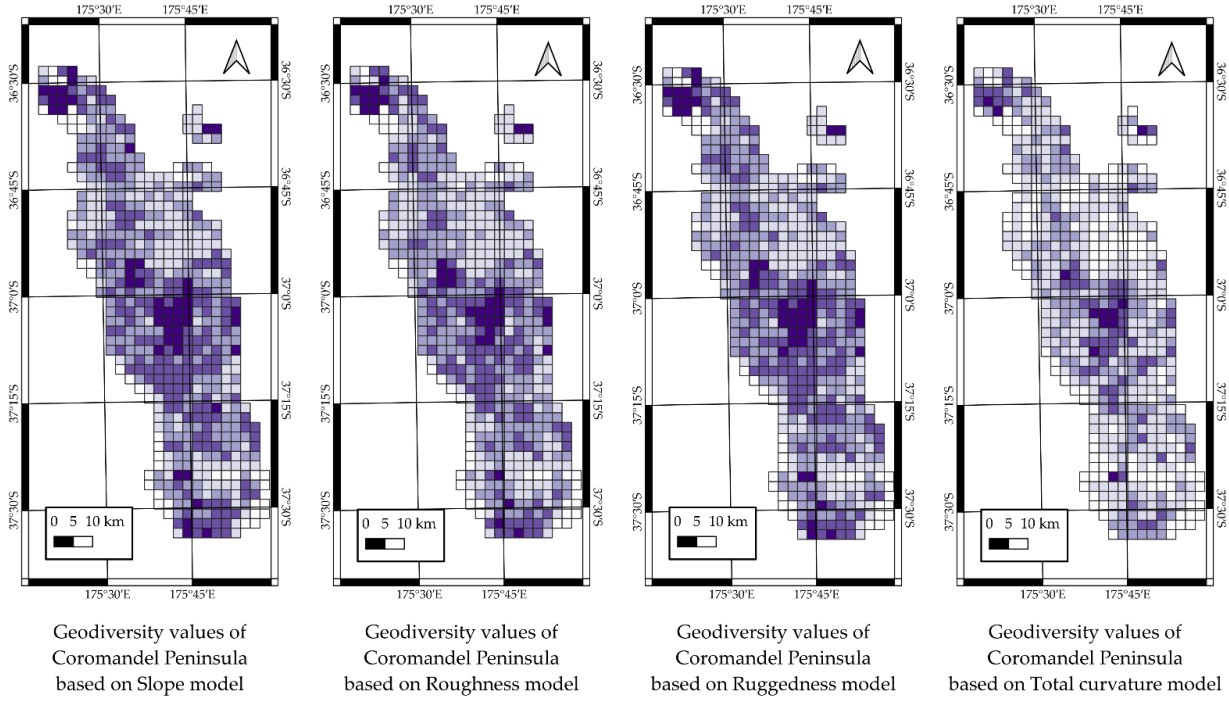

**Figure 4.** Geodiversity values of Coromandel Peninsula for geosite recognition, based on numerical geomorphological models: slope, roughness, ruggedness, and total curvature. Natural breaks (Jenks) mode of evaluation.

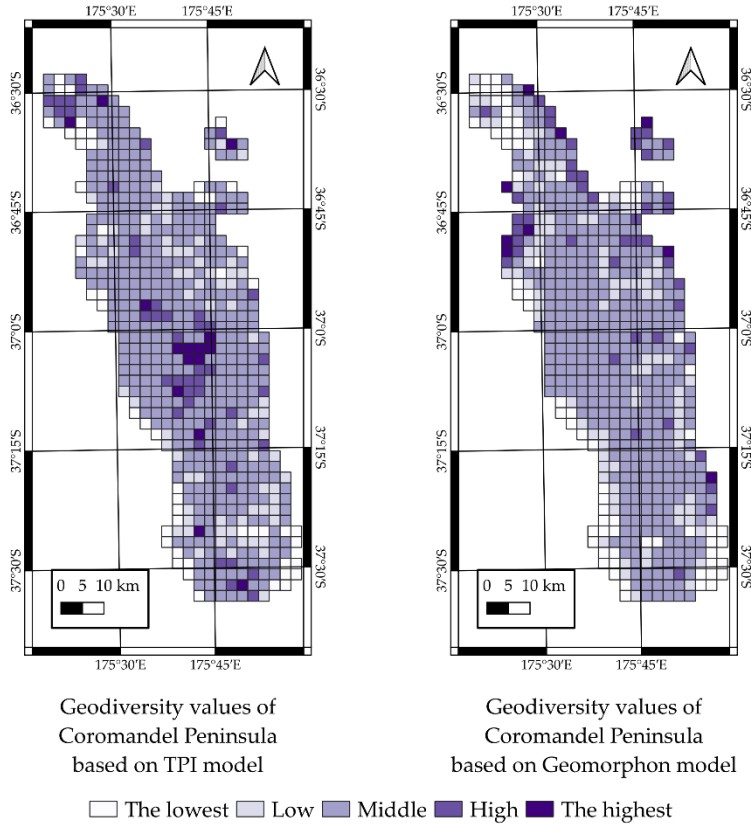

Geodiversity values of Coromandel Peninsula based on TPI model

Geodiversity values of Coromandel Peninsula based on Geomorphon model

☐ The lowest ☐ Low ■ Middle ■ High ■ The highest

**Figure 5.** Geodiversity values of Coromandel Peninsula for geosite recognition, based on landform geomorphological models: TPI and Geomorphon. Equal interval mode of evaluation.

The results for the roughness assessment show similar patterns to the slope model, but with a lower number of areas for potential geosites. Meanwhile, nearly ~25% of the north–central region displays low values, while the slope model gives values in the middle range. Nonetheless, broadly speaking, both models assign the highest values to the same regions. Additionally, the ruggedness model shows very similar results to the roughness model, with differences mostly seen in the central and in the northern parts of the region, with some values lower compared to the former model in this sub-section. The last of the four numerical models is total curvature, which results in the low and the lowest values of geodiversity for geosite recognition throughout the whole region of research. Using this model, only a few areas as defined by previous models are suggested to be treated as areas containing potential geosites.

The Topographical Position Index (TPI) is one of the landform models we calculated in this research which, compared to previous numerical models, produces values that contain information on some specific form of the landscape. These could be valleys, cliffs, hills, or mountain ranges. This results in a homogeneous pattern throughout the Coromandel Peninsula assigned a middle value, except for the north and central regions, which can still be considered high- and the highest-value places. Meanwhile, the eastern part contains only five separated areas with high value. The Mercury Islands do show the same higher-value areas as other models.

The second landform model we evaluated is Geomorphon, which contrasts with the other models in the way it describes landscapes. We created this specific evaluation system to demonstrate which type of landform could be considered more valuable in the context of this exercise. The results derived from Geomorphon are displayed as homogeneous middle values throughout our study area, same as the results from the TPI. However, unlike TPI, which still defines high-value areas as in previous models (slope, roughness, ruggedness, and total curvature), Geomorphon defines completely different areas for potential geosites

than other models. Most of these areas are in the eastern coastal areas and specific areas of the central–west region, which were not highlighted by any of the previous models.

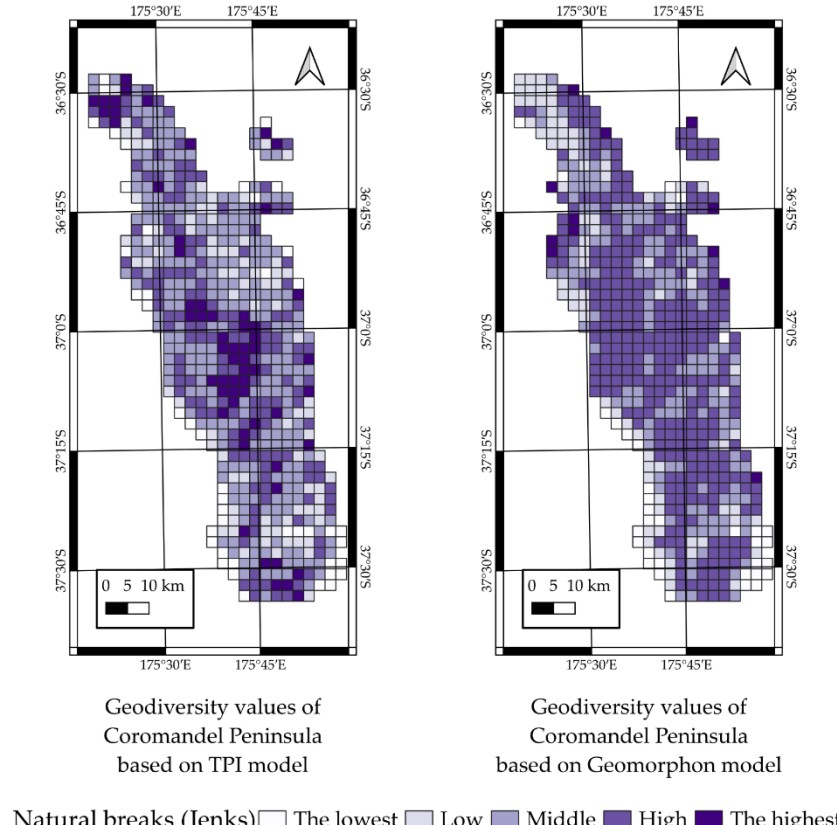

Geodiversity values of Coromandel Peninsula based on TPI model

Geodiversity values of Coromandel Peninsula based on Geomorphon model

Natural breaks (Jenks) ☐ The lowest ☐ Low ☐ Middle ☐ High ☐ The highest

**Figure 6.** Geodiversity values of Coromandel Peninsula for geosite recognition, based on landform geomorphological models: TPI and Geomorphon. Natural breaks (Jenks) mode of evaluation.

For a more accurate representation of the differences in the models, we applied an equal interval value mode calculation and created a table showing the percentages of differences between models (Table 3). The slope and Geomorphon models are similar to each other and to roughness, with similarities at around ~50%. Meanwhile, the TPI model has around 60% similarities with other models. Then, ruggedness and roughness display the most similar results at 91.1%. Similarities between total curvature and other models are lower at 23%.

**Table 3.** Comparison of results of geodiversity assessment based on different geomorphological models evaluated with equal interval.

| Similarities (%) | Roughness | Ruggedness | Total Curvature | TPI | Geomorphon |
|---|---|---|---|---|---|
| Slope | 58.2 | 54.2 | 8.6 | 65.7 | 49.5 |
| Geomorphon | 54.0 | 54.0 | 22.9 | 57.9 | |
| TPI | 62.4 | 61.4 | 11.7 | | |
| Total curvature | 18.2 | 20.1 | | | |
| Ruggedness | 91.4 | | | | |

Additionally, we applied natural breaks (Jenks) mode for comparison of results of the same models. This mode was considered as its divides the models' data based on common patterns. The slope model mostly defines the same areas, as we saw when applying the equal interval mode; however, clusters with high values are much wider, and more areas

show the highest value for geodiversity (Figure 4). Clusters became more connected to each other when covering a large area of research with high values; however, significant locations are still found in the same areas with higher geodiversity values compared to equal interval mode. In this case, we see more convergence towards the slope (natural breaks) model. In locations with the highest values in the central region of Coromandel Peninsula, we see differing values for ruggedness and roughness. Total curvature trends towards higher diversity compared to equal interval mode but still displays low values for geodiversity, especially in the central–east and southern regions of our study area.

For landform models, natural breaks mode significantly influences the resulting patterns (Figure 6). TPI shows similarities to previously described models with natural breaks mode of evaluation, in particular the northern region of the peninsula containing the highest value for "*Coromandel Granite*", and the highest values for andesite formations located at coastal areas. Additionally, some areas in the north- and central–western regions of the Coromandel Peninsula are represented by high values. Two main clusters with the highest values are recognizable with other models except Geomorphon, but with wider areas defined through application of natural breaks mode. Southern areas of the region show convergence with other models while maintaining the same patterns. Furthermore, the central–eastern region contains more areas with the highest values compared to previous models.

Geomorphon evaluated by natural breaks has a more diverse result compared to equal interval. However, around 60–70% of the region is showing high values, a significant difference from all previous models. In the northern region of the peninsula, the east coast contains high-value clusters spreading inland (Figure 6). Meanwhile, only a few places have been as assigned the highest values, two in the northeast region of the peninsula, one region in Mercury Islands (a different cell than other models), three more in the central–eastern region, and one in the southeast. Additionally, a few clusters have been highlighted in the central–western region, in contrast to other models.

For a more accurate representation of differences in the models divided by natural breaks mode, we created a table with percentages representing comparisons between models (Table 4). The slope, roughness, and ruggedness models are incredibly similar to each other at more than 84 %. Meanwhile, the TPI model has similarities to the former three at ~50%, and very low similarities with total curvature and Geomorphon. Furthermore, Geomorphon and total curvature display the lowest similarities with all other models, at less than 42%.

**Table 4.** Comparison of results of geodiversity assessment based on different geomorphological models evaluated with natural breaks.

| Similarities (%) | Roughness | Ruggedness | Total Curvature | TPI | Geomorphon |
|---|---|---|---|---|---|
| Slope | 84.3 | 93.4 | 33.2 | 56.6 | 37.4 |
| Geomorphon | 34.6 | 34.6 | 24.1 | 38.6 | |
| TPI | 52.1 | 58.4 | 19.1 | | |
| Total curvature | 41.3 | 36.4 | | | |
| Ruggedness | 87.1 | | | | |

## 4. Discussion

The aim of this research is to compare six different geomorphological models based on geological data to highlight areas potentially containing geosites and areas for further observation. The Coromandel Peninsula was chosen for modeling and testing calculations. Our calculations show that slope, roughness, and ruggedness models evaluated using the equal interval mode define similar locations with high and the highest value of geodiversity. Slight differences are seen when we use the TPI and total curvature, while we demonstrated that the Geomorphon model shows mostly homogeneous results, so we consider it to be

unsuitable for assessment of geodiversity. However, a more precise comparison shows that most of the models have similarities of more than 50%, except total curvature, which is 23% or less compared to the others. Additionally, natural breaks (Jenks) mode was utilized to examine the model's evaluation, where slope, roughness, and ruggedness have similarities in results that are higher than 84%. The TPI model shows ~55% similarities with slope and roughness models. Hence, results are dependent on which mode we used to evaluate the results of the models. Results show that models based on slope, roughness, or ruggedness are mostly exchangeable, giving similar results for geosite recognition and highlighting the same areas of interests. Meanwhile, the TPI model has fewer similarities to the slope, roughness, and ruggedness models but shows the same pattern of clusters of locations with the highest values. We demonstrated that total curvature and Geomorphon are not useful for geosite recognition, with the former showing some places with the highest values but missing others, while Geomorphon presents completely different and very homogeneous values throughout the study area. However, Geomorphon may provide higher accuracy at a lower scale of assessment without using the grid system. Slope, roughness, and ruggedness are more less exchangeable models and, together with TPI, recommended for qualitative–quantitative assessment of geodiversity for geosite recognition.

We identify the main issues with qualitative–quantitative assessment of geodiversity to be the scale of research, quality of accessible data, and the evaluation system. Our research utilized a grid of 6.25 km$^2$ cells grid to divide the Coromandel Peninsula into smaller and more convenient areas of focus. Previous research undertaken on the islands of Samoa demonstrates a more accurate non-grid system [13]; however, we do not consider this suitable for our large and diverse study area of the Coromandel Peninsula. Therefore, we instead used a grid system, which is suitable for field observation, and comparable with standard New Zealand topographical maps. This allowed us to define more precisely "geodiversity hotspots" calculated with more accurate data. The next issue is the accuracy of data. For this research, we utilized SRTM data as they cover the whole planet, are easily accessible, do not incur a cost, and provide enough resolution for our assessment. However, it is still possible to miss some important information. For example, slope, ruggedness, and roughness models are assigned low values in the central–eastern region of the peninsula, although these feature many valuable cliff sites, which are not recognizable by the SRTM model but visible in DEM based on a New Zealand topographic map. The last problem is the evaluation system. In previous research, we used a global evaluation system, which was based on slope degree values ranging from 0 to 90. However, it was not suitable for this assessment and comparison of geomorphological models, as all of them utilize different parameters, which cannot be evaluated equally. Therefore, we avoided separate evaluation of each model but calculated values by multiplying them with a global geological seven-point system for numerical models (slope, ruggedness, roughness, and total curvature). However, the evaluation system was applied to Geomorphon and TPI as they represent landforms rather than some specific parameter. Our research demonstrates the utility of this calculation for assessment, as this avoids strict evaluation systems and is suitable for numerically based models. SRTM data are adequate for qualitative–quantitative assessments for geosite recognition; however, the results should be cross-checked utilizing more accurate data if the ultimate purpose is geosite recognition. Finally, a grid system should be used for recognizing specific areas that may contain potential geosites, which than can be improved by assessing these regions with a non-grid system.

To demonstrate the accuracy of our qualitative–quantitative assessments of geodiversity more objectively, we utilized data from field observations carried out in the Coromandel Peninsula as well as sites extracted from the New Zealand Geopreservation Inventory (http://www.geomarine.org.nz/NZGI/, accessed on 28 August 2022 and https://naturemaps.nz/maps/#/viewer/openlayers/484, accessed on 28 August 2022) to check alignment between high and the highest values. To achieve this, all points of observation were overlapped on each model, utilizing natural breaks mode for evaluation (Tables 5 and 6). However, not all these points should be considered as geosites as our field

observations were based on checking the whole Coromandel Peninsula. Some points we observed may only be notable for views of offshore islands or distant mountains. In contrast, New Zealand Geopreservation Inventory research contains more points based only on specific geological or geomorphological information. Therefore, as shown in the table of our field observations, 56 locations were captured by our assessment, where slope model captures 8 locations of the highest values, then 6 locations are captured by ruggedness, and 4 locations by TPI. However, this model also has 23 locations as high values, while slope and ruggedness captured only 10 and 11, respectively (Table 5). Data about scenic points downloaded from New Zealand Geopreservation Inventory contain 76 locations of interest. All of them have been included in our assessment, giving similar results to data from field observations, or rather pattern (Table 6). Once more, slope and ruggedness captured 12 and 11, respectively, for the highest value. Meanwhile TPI has 11 for the highest and 22 for high values, which is higher than slope, roughness, and ruggedness, which captured 2 for each. Both tables demonstrate that the total curvature model is unfit for this assessment as most points falling into places with middle and low values. Additionally, Geomorphon is also demonstrated as being unsuitable for our purposes due to homogeneous results, as described in the Results section, despite capturing a high number of locations. We have visualized our results in Figure 7, where geosite recognition based on TPI is presented with our field observation points and New Zealand Geopreservation Inventory sites.

**Table 5.** Comparison of results of geodiversity assessment based on different geomorphological models with location recognized through field observation.

| Field Observation Sites | Slope | Roughness | Ruggedness | Total Curvature | TPI | Geomorphon |
|---|---|---|---|---|---|---|
| 1 | 1 | 1 | 1 | 8 | 0 | 0 |
| 2 | 9 | 16 | 13 | 20 | 7 | 7 |
| 3 | 28 | 24 | 25 | 22 | 22 | 21 |
| 4 | 10 | 14 | 11 | 5 | 23 | 28 |
| 5 | 8 | 1 | 6 | 1 | 4 | 0 |
| Total | 56 | 56 | 56 | 56 | 56 | 56 |

**Table 6.** Comparison of results of geodiversity assessment based on different geomorphological models evaluated with location from New Zealand Geopreservation Inventory.

| Geopreservation Sites | Slope | Roughness | Ruggedness | Total Curvature | TPI | Geomorphon |
|---|---|---|---|---|---|---|
| 1 | 3 | 3 | 3 | 12 | 2 | 5 |
| 2 | 17 | 23 | 19 | 34 | 11 | 11 |
| 3 | 25 | 23 | 24 | 14 | 30 | 24 |
| 4 | 19 | 19 | 19 | 10 | 22 | 36 |
| 5 | 12 | 8 | 11 | 6 | 11 | 0 |
| Total | 76 | 76 | 76 | 76 | 76 | 76 |

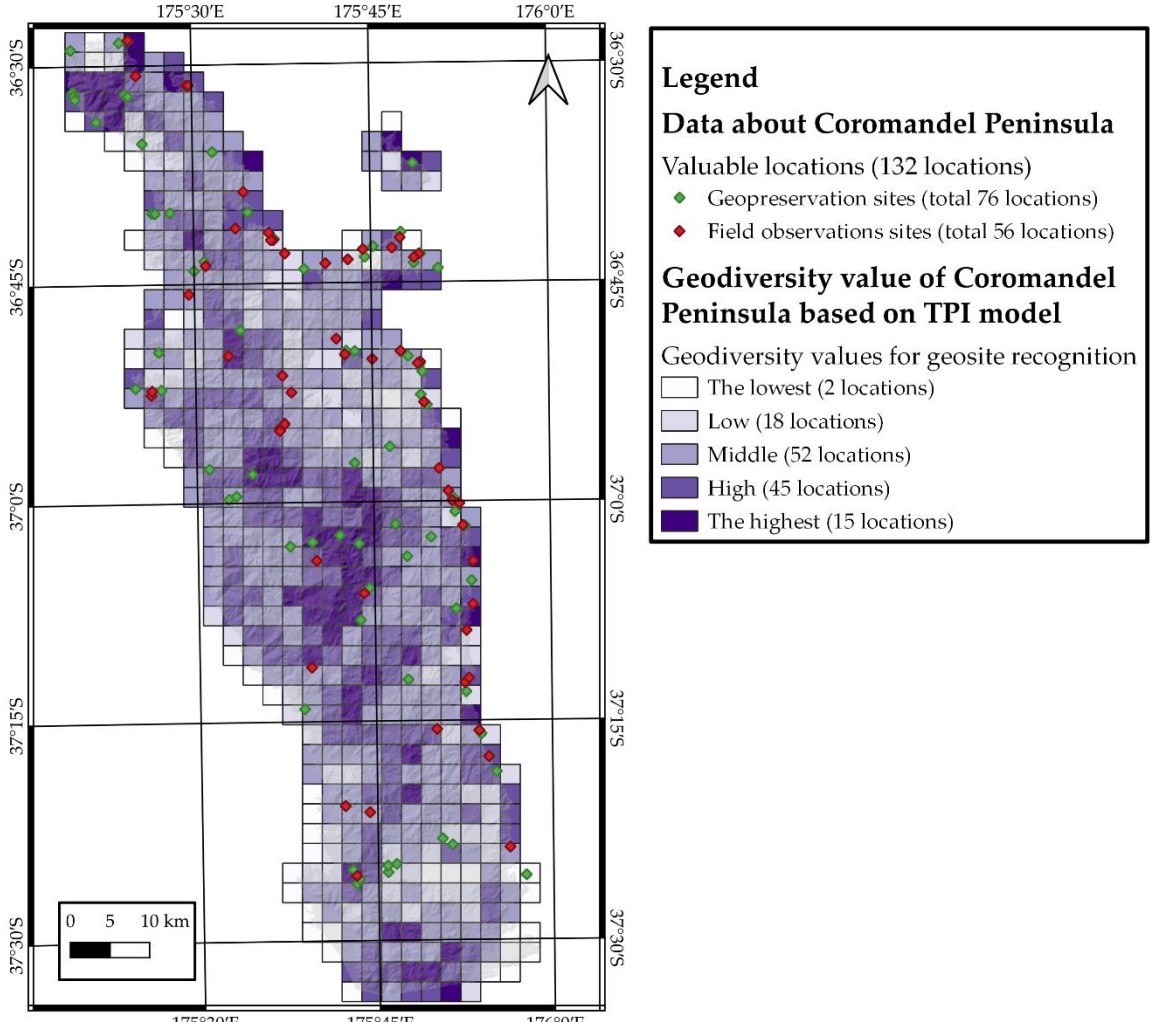

**Figure 7.** Geodiversity values of Coromandel Peninsula based on TPI model evaluated with natural breaks (Jenks) mode. Valuable locations throughout Coromandel Peninsula gathered through field observation and New Zealand Geopreservation Inventory (http://www.geomarine.org.nz/NZGI/, accessed on 28 August 2022 and https://naturemaps.nz/maps/#/viewer/openlayers/484, accessed on 28 August 2022).

For future research, the results of slope, ruggedness, and/or TPI models will be utilized to select the most valuable locations in the Coromandel Peninsula. Subsequently, we will refine our results further using more accurate data from the digital elevation model based on the topographic map of the Coromandel Peninsula. Additionally, we will create further layers of information based on abiotic nature and cultural heritage. The geodiversity description will also include knowledge about hydrology, soils, fossils, archaeological sites, minerals, etc., which have not been included in this assessment. They will be described in more detail in future research. Photographic images recorded during field observation can be used to describe the most significant geosites in the Peninsula, with more detail for geotouristic and geoeducational perspectives (Figure 8).

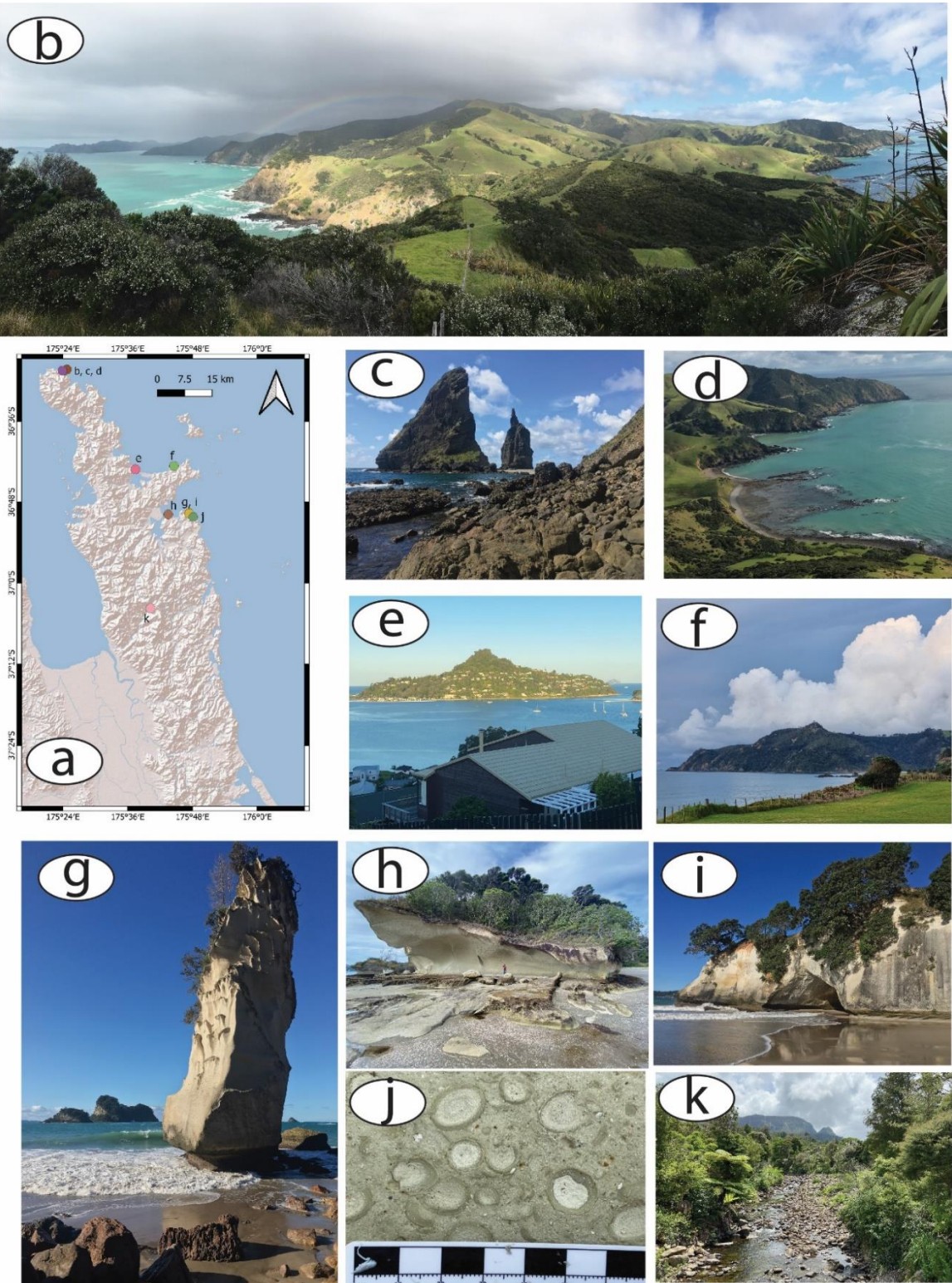

**Figure 8.** Selected geologically important and interesting sites with high geoheritage values shown on an ESRI Shaded relief map (**a**). These selected sites are compared with the geodiversity values our calculation showed. (**b**) The Fletcher Bay area in the northern Coromandel area generally falls in the high geodiversity zones; however, inland areas are more in the middle level of geodiversity values, which is consistent with the relatively simple geology and low relief of the region; (**c**) our geodiversity estimate picked up well on the local high geodiversity area of the half-section of andesitic volcanoes;

(**d**) coastal areas, especially shore platforms, are important as they commonly show well-exposed stratigraphy, such as in the Fletcher Bay; (**e**) the rhyolitic lava dome of Tairua is a key geosite that is part of a complex coastal area, and our calculation yielded a high geodiversity value for this region; (**f**) similarly, the Black Jack area showed a high geodiversity value that is consistent with its complex hydrothermal-alteration-associated geological features; (**g**) in small areas, especially in coastal areas, important local geosites were commonly missed in our estimates, which is considered to be a scale problem of the method; (**h**) in some cases, however, coastal regions composed of geologically complex features such as the Shakespeare Bay, where ignimbrite outcrops form spectacular abrasion features and perfectly exposed rocks fall within high geodiversity zones, were calculated; (**i**) the major geotouristic hot spot of the Coromandel peninsula, the Cathedral Cove, also falls within the high geodiversity field of the calculations; (**j**) small-scale features such as spectacular accretionary lapilli beds within ignimbrite deposits can be missed by our calculation, and this highlights the fact that our method should be used for first-order identification of the geodiversity elements of the region that can later be followed by detailed site exploration to locate key, normally geometrically small features; (**k**) in regions where our method provided high geodiversity values, the vegetation cover and the rugged surface commonly hinder accessibility and restrict outcrops along stream valleys, such as in the Table Mountain region along the Kauaeranga River valley.

## 5. Conclusions

Based on our assessments of geodiversity for geosite recognition, the results demonstrate that using slope, ruggedness, and roughness models produces the most similar results, which is confirmed by natural break mode for evaluating similarities ~85%. TPI is also shown to be a useful model for geodiversity recognition as its results show a similarity of ~55–60% to the former models. However, the total curvature and Geomorphon models have been demonstrated to be unsuitable for our assessment purposes due to low diversity in their results. Hence, quantitative–quantitative assessment of geodiversity for geosite recognition can be carried out with the slope, roughness, and ruggedness models, which produce nearly interchangeable results, and TPI is also suitable for this type of assessment, while Geomorphon and total curvature should be avoided.

Additional data extracted from the field observations and New Zealand data on Coromandel Peninsula show that the TPI model recognizes the highest number of areas with high and the highest values, followed by the slope and ruggedness models. In the case of roughness, despite similarities to the former models, a lower number of points are captured by the assessment. Once again, we stress the unsuitability of Geomorphon and total curvature for geosite recognition. Hence, after our additional justification of assessment accuracy, TPI can be considered one of the best models for geosite recognition utilizing our methodology, followed by slope and ruggedness.

Our assessment of geodiversity for geosite recognition demonstrates that for further observation, regions with high and the highest values must be studied at a lower scale utilizing non-grid assessment and preferably with more accurate data for elevation together with additional information about other aspects of abiotic nature. Hence, our next stage of research for geodiversity assessment of the Coromandel Peninsula will explore locations we have defined with high and the highest values to describe potential geosites more accurately with layers describing natural abiotic features to inform geotourism and geoeducation.

**Author Contributions:** Conceptualization, V.Z.; methodology, V.Z.; software, V.Z.; validation, K.N.; formal analysis, V.Z.; investigation, V.Z.; resources, K.N.; data curation, K.N.; writing—original draft preparation, V.Z.; writing—review and editing, K.N.; visualization, V.Z.; supervision, K.N.; project administration, K.N.; funding acquisition, K.N. All authors have read and agreed to the published version of the manuscript.

**Funding:** This research was funded by Massey University Post-graduate Research Scholarship granted to V.Z.

**Data Availability Statement:** Not applicable.

**Acknowledgments:** This research is part of V.Z.'s PhD research on the Coromandel Peninsula funded by the Massey University Ph.D. Scholarship. Thanks to Ilmars Gravis (The Geoconservation Trust Aotearoa Pacific) for suggested improvements to the manuscript.

**Conflicts of Interest:** The authors declare no conflict of interest.

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
