# Peer review of "Geomorphological Model Comparison for Geosites, Utilizing Qualitative–Quantitative Assessment of Geodiversity, Coromandel Peninsula, New Zealand"

_geographies, doi:10.3390/geographies2040037_

Round 1
Reviewer 1 Report
Dear authors, my corrections:
1) please add extreme geographic coordinates in all your maps
2) how you decide the extension of your study area? which are the criteria to define where the peninsula starts and finishes?
3) please check, but I think is ruggedness (not raggedness), if so, please modify in all the manuscript:
https://saga-gis.sourceforge.io/saga_tool_doc/2.2.3/ta_morphometry_16.html
4) It can be mentioned that slope and ruggedness have shown good fitting to explain different dynamic (and associated) geomorphic processes such as landslides, floods, wildfires, and paleoglacier areas, you can check and add the following references:
a) Equilibrium-line altitude and temperature reconstructions during the Last Glacial Maximum in Chirripó National Park, Costa Rica
b) Tropical glacier reconstructions during the Last Glacial Maximum in Costa Rica
c) Landslide risk index map at the municipal scale for Costa Rica
d) A geomorphometric model to determine topographic parameters controlling wildfires occurrence in tropical dry forests
e) Flood risk index development at the municipal level in Costa Rica: A methodological framework
All the best
Reviewer 2 Report
Dear Authors,
This is nice contribution extending our vision of “high-tech” opportunities for dealing with geodiversity. Although none of the approaches of this kind can be universal by definition, all of them are really important as supporting tools in geoheritage exploration. The paper is informative, novel, internationally important, and appropriately designed. Generally, it matches the standards of this kind of research. I have only some minor suggestions.
1) Abstract: please, try to compress a bit.
2) Key words: please, avoid the words used in the title.
3) Line 46: our study area – which?
4) Subsection 2.1: please, move this information to Introduction.
5) Discussion: geosites may represent too small elements like minerals, fossils, etc. What about them?
6) Lines 569-570: you mention supplementary material. Where is it? May be to include this information into the main text?
7) Please, polish the writing to avoid minor typos. For instance, Line 86: “has is”.
8) I also advise to follow better the formatting requirements of the journal.
